# The Aetiology of Tourette Syndrome and Chronic Tic Disorder in Children and Adolescents: A Comprehensive Systematic Review of Case-Control Studies

**DOI:** 10.3390/brainsci12091202

**Published:** 2022-09-06

**Authors:** Jilong Jiang, Mengxin Chen, Huifang Huang, Yanhui Chen

**Affiliations:** Department of Pediatrics, Fujian Medical University Union Hospital, Fuzhou 350001, China

**Keywords:** aetiology, risk factor, Tourette syndrome, chronic tic disorder

## Abstract

(1) Introduction: Tourette syndrome (TS) and chronic tic disorder (CTD) are common neurodevelopmental/-psychiatric disorders. The aetiological factors that contribute to the pathogenesis of TS/CTD are still poorly understood. The possible risk factors for TS/CTD are considered to be a combination of genetic, immunological, psychological and environmental factors. A comprehensive systematic review was conducted to assess the association between aetiological factors and TS/CTD. (2) Methods: Electronic databases, including PubMed, Embase, Web of Science, Wanfang data, and CNKI, were searched to identify the etiological factors of children and adolescents (≤18 years) with TS/CTD based on a case-control study. Quality assessments were performed according to the Newcastle-Ottawa scale (NOS). (3) Results: According to sample sizes and NOS values, recent evidence may support that genetic factors (*BTBD9* and *AADAC*), immunological factors (streptococcus and mycoplasma pneumoniae infections), environmental factors (conflict, history of perinatal diseases, and family history of neurological and psychiatric diseases and recurrent respiratory infections) and psychological factors (major life events) are associated with the pathogenesis of TS/CTD. (4) Conclusions: Some risk factors in different categories may be the etiological factors of TS/CTD, but there is a lack of studies on the interaction among the factors, which may require more attention in the future.

## 1. Introduction

Tourette syndrome (TS) and chronic tic disorder (CTD) are common neurodevelopmental/psychiatric disorders with an onset in childhood and adolescence, which are characterized by multiple sudden, rapid, recurrent, and non-rhythmic motor and vocal tics [1,2]. TS and CTD are among the five categories of tic disorders included in the fifth edition of the American Diagnostic and Statistical Manual of Mental Disorders (DSM-5) [3]. TS/CTD is often comorbid with obsessive-compulsive behaviour (OCB) or disorder (OCD), attention deficit hyperactivity disorder (ADHD), depression, anxiety, and other behavioural disorders [4].

However, the etiological factors that contribute to the pathogenesis of TS/CTD are still poorly understood. The pathogenesis of TS/CTD is considered to be caused by a combination of genetic, immunological, psychological and environmental factors. Although there have been some systematic reviews on risk factors of TS/CTD, many non-English studies and many types of factors have not been taken into account. To better investigate the aetiology of TS/CTD, a comprehensive systematic review of eligible studies conducted without language restriction before February 2022 was performed to assess the association between risk factors and pathogenesis of TS/CTD based on case-control studies.

## 2. Materials and Methods

This study was designed under the items of the Preferred Reporting Items for Systematic Reviews and Meta-Analyses (PRISMA) guidelines [5,6]. The study does not involve the collection of patients’ individual information; thus the approval of the Ethics Committee is not required for the study.

### 2.1. Information Sources and Research Strategies

Articles published before February 2022 were retrieved from electronic databases including PubMed, Embase, Web of Science, Wanfang data, and CNKI. The search terms for all five databases used were: (“Tourette syndrome” OR “Tourette disorder” OR “chronic tic disorder”) AND (“genetic” OR “gene” OR “immunity” OR “immune” OR “infection” OR “psychology” OR “psychosocial” OR “mental” OR “environment” OR “influencing factor” OR “risk factor” OR “etiology” OR “cause” OR “prevalence” OR “epidemiology”). The structure of the queries was tailored to the requirements of each database. The references of retrieved articles and review articles were also manually searched for studies that had been missed.

### 2.2. Inclusion and Exclusion Criteria

The following criteria were considered for the inclusion of the studies: (1) focused on risk factors for TS/CTD in children and adolescents (≤18 years); (2) case-control studies; (3) TS or CTD diagnosed according to DSM-3 [7], DSM-3TR [8], DSM-4 [9], DSM-4TR [10], or DSM-5 [3] criteria; (4) no language restrictions; (5) with or without comorbidities.

Studies were excluded as follows: (1) participants (>18 years) included or without clear data of age; (2) diagnosed according to other criteria or without clear diagnostic criteria; (3) duplicate publications of the same population; (4) abstracts, reviews, conference articles, academic dissertations, letters, editorial materials or books.; (5) participants with pediatric autoimmune neuropsychiatric disorders associated with streptococcal infections (PANDAS) or participants without TS/CTD are included in case group, and (7) studies exclusively on animal models or in vitro.

### 2.3. Assessment of Literature Quality

According to the Newcastle-Ottawa scale (NOS), studies were judged on three broad perspectives: the selection of the study groups; the comparability of the groups; and the ascertainment of either the exposure or outcome of interest for case-control or cohort studies respectively. A maximum of nine points was assigned to each study: four for selection, two for comparability, and three for outcomes. The quality of all included studies was evaluated separately by two researchers [11]. Discrepancies were discussed and adjudicated by a third researcher until consensuses were reached on every item. Studies with scores of 7~9 and 4~6 stars were defined as ‘High’ and ‘Fair’ quality, respectively, while studies with 3 or less than 3 stars were considered to represent ‘Poor’ quality [12].

## 3. Results

The process of literature retrieval and study selection is shown in Figure 1. A total of 3288 results were retrieved. After the exclusion of duplication, the publications were independently screened by two researchers according to the inclusion and exclusion criteria. Finally, thirty-three studies were selected and included in this systematic review as shown in Table 1.

### 3.1. Characteristics of Studies

The characteristics of the studies, including the first author, published year, the countries of participants, the diagnosis of cases, the number of cases and controls, the categories of factors, and the results of the included studies, are shown in Table 1. Three studies [42,43,44] could not be classified well. Several factors included in the study by Zhao et al. [33] did not fit any of the four categories.

### 3.2. Quality of Studies

The details of NOS values and quality of studies are shown in Table 1 and Appendix A.

### 3.3. Etiological Factors

#### 3.3.1. Genetic Factors

Dopaminergic system-related genes were of particular interest (6/13). Five [15,16,17,20,21] of the six studies focused on the dopaminergic system relative genes were conducted in the Chinese population. Another study [13] conducted in the Italian population recruited subjects with TS (*n* = 15) and matched normal controls (NC) (*n* = 15). The mRNA expression of dopamine D5 receptor (DRD5) in the TS group was significantly higher than in NC group (*p* < 0.001), but there were no differences in DRD 2, 3 or 4 mRNA expression. In the study by Lu et al. [21], the genotype and allele frequencies of *DRD4* 616C/G were significantly different between CTD (*n* = 84) and NC (*n* = 100) (*p* = 0.02; *p* = 0.004). However, some results of the studies were inconsistent. Ji et al. [17] also focused on *DRD4*, and they did not find differences in genotype and allele frequencies of *DRD4* exon III 48 bp variable number of tandem repeats (VNTRs). Other dopaminergic system-related genes were also investigated, but no positive results were obtained. He et al. [16] found no significant differences in the genotype and allele frequencies of *DRD3* rs6280 single nucleotide polymorphisms (SNPs) between TS (*n* = 160) and NC (*n* = 90) (*p* = 0.161; *p* = 0.423). A study [15] compared common TS (*n* = 63), refractory TS (*n* = 52) and NC (*n* = 57) to analyze the polymorphisms of dopamine transporter 1 (*DAT1*). There were no significant differences in genotype and allele frequencies of *DAT1* 40bp VNTRs among each group (*p* = 0.423). In the study by Liu et al. [13], which included 106 subjects with TS, there were no significant differences in the polymorphisms of dopamine beta-hydroxylase (*DBH*) Taq1 between TS and NC.

Forty-six probes were identified with a fold change > 1.5 in the study by Lit et al. [19] which included 30 subjects with TS and 28 normal NC, but TS could not be separated from NC by unsupervised hierarchical clustering and principal components analysis. Tian et al. [24] detected that there were significant differences in the expressions of 376 exon probe sets between TS (*n* = 26) and NC (*n* = 23) (*p* < 0.005, fold change > |1.2|). Only two exons had a corrected *p* < 0.9 by using the false discovery rate (FDR) correction for multiple comparisons, but no exon had a multiple comparison corrected *p* < 0.05. Ninety genes had different expression of a single exon (*p* < 0.005), and there were three genes with a corrected *p* < 0.05 based on the Benjamini and Hochberg FDR of <0.05 for multiple comparison correction, including ubiquitously transcribed tetratricopeptide (*UTY*), myosin ⅩⅧB (*MYO18B*), and potassium voltage-gated channel, subfamily H (eag-related), member 4 (*KCNH4*).

Yuan et al. [25] compared TS (*n* = 200) with NC (*n* = 300) and identified two variations of arylacetamide deacetylase (*AADAC*) in two unrelated patients with TS, including a heterozygous splice-site variant, c.361 + 1G > A (rs762169706), and a missense variant, c.744A > T (p.R248S, rs186388618). However, the c.744A > T variant was also identified in two controls. Lei et al. [18] did not find differential expressions of histidine decarboxylase (*HDC*), HECT domain and RCC-1 like domain 1 (*HERC1*), HECT domain and RCC-1 like domain 2 (*HERC2*), cholinergic receptor, neuronal nicotinic alpha polypeptide 7 (*CHRNA7*), ubiquitin protein ligase E3A (*UBE3A*), or ubiquitin specific peptidase 3 (*USP3*) in TS subjects (*n* = 30) compared with NC subjects (*n* = 30). While the expression of amyloid precursor protein-binding protein A2 (*APBA2*) in TS (*n* = 84) was significantly lower than in NC (*n* = 100) (*p* < 0.01), whose sample sizes was expanded. Guo et al. [14] found that the variant rs9296249 of BTB/POZ domain-containing protein 9 (*BTBD9*) was significantly correlated with TS (*n* = 100) compared with NC (*n* = 440) (*p* = 0.010). However, other variants of *BTBD 9* and serotonin 2C receptor were not significantly different between both groups (*p* > 0.05).

There were not just studies on genes, but also some studies on epigenetics. Both studies focused on microRNA (miRNA, miR), and were conducted in the Italian population. Rizzo et al. [23] identified downregulated expression of miR-429 by analyzing 52 TS patients and 15 NC patients (Wilcoxon test *p* = 0.01; *t*-test *p* = 0.004). The study by Mirabella et al. [22] compared TS (*n* = 6), TS with Arnold-Chiari syndrome (ACTS) (*n* = 11) and NC (*n* = 8). The results showed that miR-23a-3p was upregulated in TS compared to NC (1.67-fold), miR-130a-3p was downregulated in ACTS or TS compared to NC (−1.56-fold; −1.61-fold), miR-222-3p and miR-451a were upregulated in ACTS compared to NC (1.95-fold; 1.58-fold), and the FDR was <0.05 for each pairwise comparison.

Seven studies [14,15,16,19,20,22,23] were judged to be of high quality, and six studies [13,17,18,21,24,25] were judged as fair quality according to the NOS. The studies by Lei et al. [18] and Ferrari et al. [13] highlighted the defects in the selection and description of controls, and potential selection biases of cases. In additional, potential selection biases of cases were also found in six studies [14,17,19,21,23,25], and selection biases of controls were found in five studies [14,16,20,21,25]. The study by Guo et al. [14] was the only one which was blinded for ascertainment of exposure.

Both negative and positive results were reported in the studies focused on different genes, different loci, or miRNA. There is little high-quality evidence to support the association between dopaminergic system-relate genes and TS/CTD. Ferrari et al. [13] demonstrated DRD5 mRNA expression was high in TS. Polymorphisms of *DRD4* 616C/G associated with CTD were reported in the study by Lu et al. [21] Polymorphisms of *BTBD9* and lower expression of *APBA2* were reported in the studies by Guo et al. [14] and Lei et al. [18], respectively. However, among the four studies, only the Guo et al. study was of high quality. Differential expressions of several miRNAs in TS were reported in the studies by Rizzo et al. [23] and Mirabella et al. [22], and both studies were identified as high quality.

#### 3.3.2. Immunological Factors

Four studies involved Streptococcal infections. The study by Ji et al. [29] including TS (*n* = 32) and NC (*n* = 30) showed that the titers of anti-streptolysin O (ASO) in TS were significantly higher than that in NC (*p* < 0.05). Another study by Rizzo et al. [30] found anti-streptolysin titers were increased (>400 IU/L) in 41/69 (59.4%) in TS patients and 14/72 (19.4%) NC (*p* = 0.000). Dong et al. [28] reported that ASO titers were elevated (≥200 IU/L) in 27 (56.3%) TS patients and 2 (10.0%) NC patients (*p* < 0.05). Cheng et al. [27] compared TS (*n* = 67) with NC (*n* = 60) and found that ASO titers increased (≥250 IU/L) in 13/66 (19.7%) TS and 0 NC (*p* < 0.01).

Other relevant factors were also investigated. The study by Ji et al. [29] also found that the level of interleukin (IL)-6 in TS patients was higher than that in NC patients (*p* < 0.05). However, they did not find any significant differences in the value of cluster differentiation (CD) 3+, CD4+, CD8+, and CD4+/CD8+, or the level of IL-8 between both groups (*p* > 0.05). Cheng et al. [27] demonstrated that the levels of soluble IL-6 receptor (sIL-6R) and soluble glycoprotein (sgp) 130 in TS were significantly higher than in NC patients (*p* < 0.01). In their study, significant differences in positive rates of anti-brain antibody (ABAb) and antinuclear antibody (ANAb) were also reported (66% vs. 4% and 53% vs. 25%, *p* < 0.01 respectively). T Rizzo et al. [30] also identified that 22 (31.9%) TS patients had positive anti-basal ganglia antibodies, while 7 (9.7%) NC patients had the positive antibodies (*p* = 0.002). Mycoplasma pneumoniae (MP) was another pathogen of interest in the included studies, in addition to streptococcus. The results of the study by Chang et al. [26] showed that the positive rates of MP antibody (antibody titers ≥ 1:160) and MP-specific antibody immunoglobulin (Ig) A were significantly higher in TS (*n* = 60) than in NC patients (*n* = 60) (30.0% vs. 0%, *p* = 0.0012; 35.0% vs. 3.3%, *p* = 0.0031), but there were no significant differences in MP-specific antibody IgM or IgG (*p* = 0.374; *p* = 0.56). Yang et al. [31] identified that there were significant differences in the positive rates of antibody and IgA of MP in TS (*n* = 50) compared to NC (*n* = 50) (32.0% vs. 2.0%, *p* = 0.001; 30.0% vs. 4.0%, *p* = 0.003). Similarly, no significant differences in MP-specific antibody IgM or IgG were found (*p* = 0.36; *p* = 0.54). The study by Yuce et al. [32] included TS only (*n* = 19), OCD plus TS (*n* = 13) and controls (*n* = 35), and revealed that the presence of any allergic disease, positive skin prick test rates and rates of eczema diagnosis were significantly higher in the TS plus OCD group compared with control group (*p* = 0.018; *p* = 0.011; *p* = 0.001). However, there were no significant differences in these factors, as well as asthma diagnoses, in the levels of IgE or eosinophil counts between TS only group and control group (*p* = 0.076; *p* = 0.607; *p* = 0.294; *p* = 0.480; *p* = 0.554; *p* = 0.539). Frequencies of recurrent respiratory tract infections within one year were reported to be correlated with TS in the study by Zhao et al. [23] (*p* < 0.01, OR 1.040, 95%CI 1.019~1.062).

Quality assessments resulted in an NOS standard of ‘high’ for the study by Rizzo et al. [30] and ‘fair’ for the other seven studies [26,27,28,29,31,32,33]. Comparability and exposure ascertainment were inadequate in the seven studies. The study by Cheng et al. [27] had loss of one subject in the results section without description. With the exception of the study by Cheng et al. [27], all other studies had obvious defects in control selection.

No study included CTD subjects. Although there was only one high-quality study, and different reference indicators were applied in some studies, the results of four studies were consistent that positive ASO or anti-streptolysin was associated with TS. This indicated that streptococcal infection plays a role in the pathogenesis of TS. It was reported that the positive rates of MP antibody and special antibody IgA in TS were increased, especially in TS with OCD comorbidity, in two ‘fair’ studies. The evidence for elevated IL-6 ABAb and ANAb in TS was ‘fair’. There was no available evidence to prove that IL-8, IgE, and T lymphocytes were associated with TS/CTD.

#### 3.3.3. Environmental Factors

Family relevant factors and perinatal factors were used to assess environmental factors. Gu et al. [34] analyzed the family environment of TS (*n* = 60) and NC (*n* = 60). The results showed that there were low intimacies and high conflicts in TS (*p* = 0.000; *p* = 0.000), but there were no significant differences in emotional expression and independence (*p* = 0.87; *p* = 0.103). Liu et al. [38] compared the TS only group (*n* = 40), and the TS plus OCD group (*n* = 33) with NC group (*n* = 40) to analyze family environment and parenting style. Conflicts, father’s refusal and denial, father’s overprotection, and mother’s over-intervention and overprotection were higher in TS only group and TS plus OCD group (ANOVA: *p* = 0.000; *p* = 0.003; *p* = 0.000; *p* = 0.001), and father’s emotional warmth was lower in TS only group and TS plus OCD group (ANOVA: *p* = 0.004). In addition, organization was lower in the TS only group (*p* = 0.035). No other factors were shown to be associated with TS in this study. Liu et al. [45] demonstrated that five family environmental factors (including conflicts, recreational orientation, independence, organization and control) in TS (*n* = 55) were significantly different from those in matched controls (*n* = 55) (*p* < 0.01).

Prenatal and perinatal factors, as well as other factors, were also detected. Motlagh et al. [40] investigated the risk factors of prenatal and perinatal periods, including heavy maternal smoking, high maternal stress levels, medical condition, and low birth weight. There were trends that the frequencies of heavy maternal smoking and the levels of severe maternal psychosocial stress during pregnancy were higher in TS only (*n* = 45) and TS plus ADHD (*n* = 60) compared with NC (*n* = 65), but without significant differences (heavy smoking: *p* = 0.19, *p* = 0.052; severe psychological stress: *p* = 0.11, *p* = 0.07). Klug et al. [36] analyzed five parental variables (including parents’ age, parents’ education, and marital status), five prenatal variables (including prior pregnancy termination, and live birth now dead), five perinatal variables (including Apgar scores, and birthweight) in TS (*n* = 92) with NC (*n* = 460). Prior pregnancy termination was the only significant risk marker (*p* < 0.05), while other variables were not significantly different. The study by Zhao et al. [33] assessed other 18 risk factors except recurrent respiratory tract infections within one year, with nine related factors involved. There were significant differences in several factors between TS (*n* = 206) and NC (*n* = 125), including history of perinatal diseases, and family history of neurological and psychiatric diseases (single-factor analysis: *p* < 0.001, *p* < 0.001; multiple-factors analysis: *p* = 0.065, *p* = 0.547, *p* < 0.001, *p* < 0.001). There were significant differences in parents’ personality traits and parents’ lifestyle by single-factor analysis (*p* = 0.002; *p* = 0.016; *p* = 0.002), while there were no significant differences in these factors by multiple-factors analysis (*p* = 0.065; *p* = 0.547; *p* = 0.054). No statistical differences were found in other points. Six relevant factors were explored in the study by Li et al. [35] Compared with control group (*n* = 100), threatened abortion, birth injuries, food preference, exposure of games and television, strict discipline, and anorexia were related risk factors for the TS group (*n* = 160) by multiple-factors analysis (*p* = 0.23; *p* = 0.000; *p* = 0.012; *p* = 0.005). Khalifa et al. [35] investigated TS (*n* = 25) and controls (*n* = 25). The percentage of first-degree relatives with psychiatric disorders in TS patients was higher than that in controls (80% vs. 20%, *p* < 0.001). TS mothers were twice as likely to have pregnancy complications and were younger than control mothers when giving birth to the index child (*p* < 0.001). There were no differences in education of parents, socioeconomic status or divorce rate between the two groups.

Studies by Motlagh et al. [40], Gu et al. [34] and Liu et al. [38] were considered ‘high’ quality. These studies performed well in terms of selection, but poorly in ascertainment of exposure and comparability. The other five studies were considered ‘fair’ quality. Exposure ascertainment, control selection, and comparability were poor. For some reason, two controls did not receive the same interview and examination in the study by Khalifa et al. [35] The control group was not well defined in the study by Li et al. [37]

There were both consistent and inconsistent results in the three studies of family environmental factors. Conflict was consistently identified as a risk factor for TS. Several aspects of parenting were recognized to be related to TS. Although Liu et al. [45] found that control and independence were related to TS, the other two studies did not support this. The sample size and quality of the study by Liu et al. [45] were smaller than those of the other two studies, and the grouping of the study by Liu et al. [45] was less precise than that of another study [38]. These factors may not be so important for TS. The results of perinatal factors were inconsistent, and the only study with high-quality did not identify any factor that played a role in TS/CTD.

#### 3.3.4. Psychological Factors

Horesh et al. [39] analyzed TS (*n* = 41) and NC (*n* = 24) by using a life experience survey and a junior temperament and character inventory. They failed to prove the association between TS and stressful life events without any significantly different variable. Another study by Horesh et al. [41] including TS (*n* = 132) and NC (*n* = 49), identified there was a higher quantity of negative major life events and a lower quantity of positive major life events in TS (*p* < 0.05; *p* < 0.05). However, minor life events were not related to TS.

Both studies were considered ‘fair’. Both of them had similar flaws in comparability and exposure ascertainment. Both studies were designed to analyze the effects of life events on TS but the results were inconsistent. The study by Horesh et al. [41] supported the association between TS and major life events, which was more recent, and enrolled more cases and controls. TS may be more or less related to life events.

#### 3.3.5. Other Factors

Liu et al. [44] tested TS (*n* = 22) and NC (*n* = 18) and showed that levels of plasma prolactin increased in TS (0.01 < *p* < 0.05). Erbay et al. [43] investigated three kinds of hormones, including testosterone, dehydroepiandrosterone sulfate (DHEA-S) and cortisol of TS/CTD (*n* = 26) and NC (*n* = 25). Higher levels of testosterone and DHEA-S were found in TS/CTD compared to NC (*p* = 0.019; *p* = 0.025), but no statistical differences were found between the cortisol levels in the two groups (*p* = 0.642). Corbett et al. [42] estimated the reactivity and diurnal rhythms of cortisol in TS (*n* = 20) and neurotypical controls (*n* = 16). The outcomes showed that the diurnal pattern was not different, but lower cortisol levels in the evening were observed (*p* = 0.08). Thus, to assess the response to stress, four timing periods (arrival, post-mock, pre-MRI, and post-MRI) were studied, and higher levels of cortisol caused by the MRI environment in TS were revealed (*p* = 0.033). The study by Zhao et al. [33] did not find differences in the levels of lead, selenium, zinc, or 25-hydroxyvitamin D (*p* = 0.085; *p* = 0.24; *p* = 0.829; *p* = 0.361).

The study by Erbay et al. [43] was judged to be of fair quality. Studies by Liu et al. [44] and Corbett et al. [42] were both judged as ‘fair’ based on NOS standards. The study by Liu et al. [44] did not meet the standards of NOS well, probably because it was published before the NOS was available. The study by Corbett et al. [42] was low in terms of comparability, exposure ascertainment, and case representativeness.

Cortisol levels were tested in two studies, but the study by Erbay et al. [43] only evaluated the levels of cortisol at 9:00 in the morning. Neither study found differences in cortisol levels in the morning, and evidence did not support a link between cortisol levels in the morning and TS. Decreased levels in the evening and elevated levels in response to stress were reported, but the sample size included in the study was small, and the quality of the study was fair. Increased plasma prolactin has been reported but similar shortcomings existed in small sample size and non-high quality of the study. Microelements or 25-hydroxyvitamin were not identified to be related to TS in a ‘fair’ study.

## 4. General Discussion

The majority of studies included had similar flaws with control sources, which might affect the validity of the results. Among the nine studies, the sample size of TS/CTD patients was greater than 100, but only four studies, including two studies of genetic factors, and two studies of environmental factors, involved sample sizes of both TS/CTD patients and controls greater than 100. In addition, the sample sizes of TS/CTD patients in seven studies were smaller than 30, which included three studies of high quality. According to sample sizes and NOS values, our review may be more supportive of that *BTBD9*, *AADAC*, streptococcus and MP infections, conflict, history of perinatal diseases, family history of neurological and psychiatric diseases, recurrent respiratory infections, positive and negative major life events being associated with the pathogenesis of TS/CTD.

*BTBD9* and *AADAC* may be associated with TS/CTD according to studies in which the sample sizes of both TS/CTD patients and controls were greater than 100. The relationship between *BTBD9* and TS has been reported in the study by Rivière et al. [46] *BTBD9* mutated mice exhibited alterations in circling behaviour and more restlessness, which are regarded as tic-like behaviours [47]. *BTBD9* knockout (KO) mice have increased striatal neural activity [48]. The striatum, as one of the components of the cortico-striato-thalamo-cortica (CSTC) circuit, plays an important role in the pathogenesis of TS/CTD [5]. The study by Pagliaroli et al. [49] also reported the association between *AADAC* and TS. Moreover, the association of *AADAC* with TS was previously identified by a meta-analysis with a large sample size [50]. The expression of *AADAC* has been detected in the brain regions of humans and mice and been related to the pathogenesis of TS/CTD [50,51]. These findings suggest the possibility of *BTBD9* and *AADAC* as etiological factors of TS/CTD. Effective drugs for TS/CTD therapy, such as haloperidol, aripiprazole and ecopipam mainly act on DRD2 and DRD1 [52,53,54], while differential expression of DRD2 mRNA was not found in TS/CTD in the study by Ferrari et al. [13] The roles of related genes in the pathogenesis of TS/CTD are still unclear and need further study. This may help determine the pathogenesis of TS/CTD and find new treatment options

Two studies focused on MP infections in TS, and both studies were aimed at the Chinese population. It was reported that macrolides effectively treated one case of TS infected by MP [55]. Moreover, Müller et al. considered MP infection as an aggravating factor in TS [56]. Our study is basically consistent with the previous meta-analysis by Lamothe et al. [57] in that there are an elevated level of ASO in TS. Rizzo et al. [30] identified that elevated anti-streptolysin titers are accompanied by positive anti-basal ganglia. However, different ASO titer cut-off values were used in these studies, which may bias our results. This suggests, at least in part, that MP and streptolysin infections are related to TS/CTD, but the mechanism by which infections cause TS/CTD remains unclear. More studies are needed to focus on the mechanism of TS/CTD caused by infections.

Chao et al. [58] performed a systematic review of prenatal factors for TS, and found that maternal smoking and low birth weight are associated with TS. This is inconsistent with our results, and may be due to the different types of literature included in our study. Family environmental factors, including conflict and family history of neurological and psychiatric diseases may be linked to TS/CTD. Comprehensive family-based behavioural interventions have been implemented for TS/CTD [59]. A family history of neurological and psychiatric diseases has been focused on, but a combination of genetic and environmental factors could be involved. Future studies may need to pay more on attention to the interaction of genetic and non-genetic factors.

Life events reflect psychological stress, and psychological components may affect immunity and health [60]. Two studies by the same first author involved life events with different consequences, and in the later published report, the sample sizes of both TS patients and controls were greater than 100. Personality patterns are factors that regulate the correlation between genetic susceptibility, life events, and the pathogenesis of a psychiatric disorder [61]. Psychological interventions are effective on TS/CTD [62] and they are listed as first-line treatments for TS/CTD [63]. This suggests that positive and negative major life events are the underlying etiologial factors of TS/CTD.

TS and CTD are complex and heterogeneous diseases and their pathogenesis may be multifactorial [5]. In our review, we found that TS/CTD might be related to many factors. However, no study was found to focus on the interaction of genetic and non-genetic factors, and these should be further studied.

Our systemic review is based on several factors. Heterogeneities in comorbidities, sample sizes, patient characteristics, sources of controls and cases, and ascertainment of exposure may confound comparisons between etiological factors and between studies. Some studies did not provide a clear number of females or males, so reviewer reliability is difficult to judge. Some factors (genetic factors, environmental factors) were estimated more in adults or based on family studies or cohort studies. Our review was limited to the age ≤ 18 years based on case-control studies, which may have led to failure in capturing more relevant available data.

## 5. Conclusions

Our systematic review found recent evidence to support that genetic factors (*BTBD9* and *AADAC*), immunological factors (streptococcus and MP infections), environmental factors (conflict, history of perinatal diseases, and family history of neurological, and psychiatric diseases and recurrent respiratory infections) and psychological factors (positive and negative major life events) are associated with the pathogenesis of TS/CTD. There is still a lack of studies on the interaction between these factors. Further studies are needed to take into account their interaction to understand the causative factors of TS/CTD.

## Figures and Tables

**Figure 1 brainsci-12-01202-f001:**
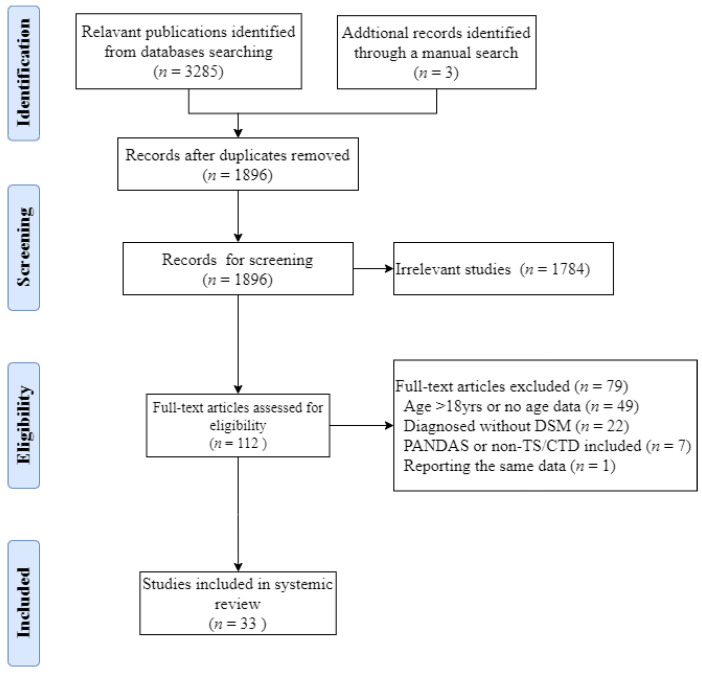
PRISMA flow diagram of study selection.

**Table 1 brainsci-12-01202-t001:** Characteristics of the included studies.

Results	Study Quality	NOS Values	Categories	Cases/Controls (N)	Diagnosis	Country	Studies
Higher DRD5 mRNA levels in TS compared to healthy controls. No differences in DRD2, DRD3, or DRD4. mRNA.	Fair	4	genetic	15/15	TS	Italy	[13]
A significant association between TS and the variant rs9296249 of *BTBD9*. No differences in the other four variants.	High	7	genetic	110/440	TS	China	[14]
No significant differences in polymorphisms of *DAT1* between groups.	High	8	genetic	115/57	TS	China	[15]
No statistical differences in the allele and genotype frequencies of *DRD3* rs6280 SNPs between TS and controls.	High	7	genetic	160/90	TS	China	[16]
No significant in *DRD4* exon III 48 bp variable number of tandem repeats.	Fair	6	genetic	86/51	TS	China	[17]
Significant differences in *APBA2* expression between TS and NC.	Fair	5	genetic	84/100	TS	China	[18]
No global expression differences between TS and controls. Within each age strata (5–9, 10–12, and 13–16), the expression of many genes differed between TS and controls.	High	7	genetic	30/28	TS	the United States	[19]
No significant differences in the polymorphism of *DBH* Taq1 digestion between TS and NC.	High	7	genetic	106/80	CTD	China	[20]
Significant differences in both genotype and allele frequencies of *DRD4* 616C/G between CTD and controls.	Fair	4	genetic	85/100	CTD	China	[21]
miR-23a-3p upregulated in TS compared to NC. miR-130a-3p downregulated in ACTS and TS compared to NC. miR-222-3p and miR-451a upregulated in ACTS compared to NC.	High	7	genetic	17/8	TS	Italy	[22]
MiR-429 significantly downregulated in TS to normal controls.	High	7	genetic	58/28	TS	Italy	[23]
Expressions of 376 exon probe set significantly different between TS and NC, but no exon with multiple comparisons corrected *p* < 0.05. 90 genes (transcripts) differently expressed of one exon in TS compared to NC, while three genes with a corrected *p* < 0.05 based on the Benjamini and Hochberg FDR for multiple comparison correction.	Fair	6	genetic	26/23	TS	the United States	[24]
Two variants of *AADAC*, including c.361 + 1G > A, and c.744A > T, identified in two unrelated TS patients. The c.361 + 1G > A variant absent in controls, the c.744A > T variant identified in two NC.	Fair	6	genetic	200/300	TS	China	[25]
Significantly higher MP antibody (titers ≥ 1:160) and MP-specific antibody IgA in TS than in NC.	Fair	6	immunological	60/60	TS	China	[26]
ASO titers raised (≥250 IU/L) in 13/66 (19.7%) TS and 0 NC.	Fair	4	immunological	67/64	TS	China	[27]
ASO titers raised (≥200 IU/L) in 27/48 (56.3%) TS and 2/20 (10.0%) NC.	Fair	6	immunological	48/20	TS	China	[28]
Significantly higher titers of ASO in TS than in NC.	Fair	6	immunological	32/30	TS	China	[29]
Raised anti-streptolysin titers in 41 of 69 (59%) TS and 14 of 72 (19%) controls. Positive anti-basal ganglia antibodies in 22 of 69 (32%) TS compared with 7 of 72 (10%) controls.	High	7	immunological,	69/72	TS	Italy	[30]
Significant differences in the positive rates of MP antibody and MP-specific antibody IgA in TS compared to NC.	Fair	6	immunological	50/50	TS	China	[31]
More comorbid allergic diseases in TS compared to controls. No significant differences in IgE levels and eosinophil counts.	Fair	6	immunological	25/25	TS	Turkey	[32]
Significant differences in frequencies of recurrent respiratory infections within one year, history of perinatal diseases, and family history of neurological and psychiatric diseases. No significant differences in the other variables.	Fair	6	Immunological, environmental	206/125	TS	China	[33]
Low intimacy and high conflict in TS. No significant differences in emotional expression or independence.	High	7	environmental	60/60	TS	China	[34]
First-degree relative with psychiatric disorders in eighty per cent of TS. Non-significant of reduced optimality score in the prenatal, perinatal or neonatal periods in TS compared to controls. No differences in socio-economic status.	Fair	5	environmental	25/25	TS, CTD	Sweden	[35]
Significant differences in prior terminations between TS and controls.	High	7	environmental	72/460	TS	the United States	[36]
According to multiple-factors analysis, threatened abortion and birth injuries, food preference, exposure to games and television, strict discipline, and anorexia related to TS.	Fair	5	environmental	160/100	TS	China	[37]
Higher conflict, rejection, and denial from father, overprotection from father, over-intervention and overprotection from mother, and lower emotional warmth from father in both TS only and TS plus OCD.	High	7	environmental	73/40	TS	China	[38]
Five (conflict, recreational orientation, independence, organization, and control) over ten family environmental factors of TS significantly different from those of matched controls.	Fair	6	environmental	55/55	TS	China	[39]
Higher frequencies of heavy maternal smoking and the levels of severe maternal psychosocial stress during pregnancy in TS only and TS plus ADHD compared to NC, but without significant differences.	High	7	environmental	105/65	TS	the United States	[40]
No significant differences in life events between TS and NC.	Fair	6	psychological	41/24	TS	Israel	[39]
Major life events correlated with TS. Minor life events correlated with more severe symptomatology.	Fair	6	psychological	132/49	TS	Israel	[41]
Lower evening cortisol for TS. Higher levels of cortisol in response to MRI environment for TS.	Fair	6	other	20/16	TS	the United States	[42]
Higher testosterone and DHEA-S levels in TS than in controls. No statistical differences between the cortisol levels.	Fair	6	other	26/25	TS, CTD	Turkey	[43]
Significantly higher soluble IL-6 receptor and soluble glycoprotein 130 in TS than in NC.	Fair	4	other	22/18	TS	China	[44]

AADAC: arylacetamide deacetylase; ACTS: Tourette syndrome comorbidity with Arnold-Chiari syndrome; ADHD: attention deficit hyperactivity disorder; APBA2: amyloid precursor protein-binding protein A2; ASO: anti-streptolysin O; CTD: chronic tic disorder; DAT1: dopamine transporter 1; DBH: dopamine beta-hydroxylase; DRD2: dopamine D2 receptor; DRD3: dopamine D3 receptor; DRD4: dopamine D4 receptor; DRD5: dopamine D5 receptor; FDR: false discovery rate; IgA: immunoglobulin A; IgE: immunoglobulin E; IL-6: interleukin-6; miR: microRNA; MP: Mycoplasma pneumoniae; N: number; NC: normal controls; NOS: Newcastle-Ottawa scale; OCD: obsessive-compulsive disorder; SNPs: single nucleotide polymorphisms; TS: Tourette syndrome.

## Data Availability

Not applicable.

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
