# Peer review of "The Aetiology of Tourette Syndrome and Chronic Tic Disorder in Children and Adolescents: A Comprehensive Systematic Review of Case-Control Studies"

_brainsci, 2022, doi:10.3390/brainsci12091202_

Round 1

Reviewer 1 Report

Overview

Jiang et al present a review of case control studies focused on identifying genetic and non-genetic factors associated with the pathogenesis of Tourette syndrome (TS) and chronic tic disorder (CTD). The authors provide a summary of studies that also reports quality measures for each study. Although it is comprehensive, it does not clearly present a meaningful synthesis of results across studies or provide suggestions for future research other than the need for larger sample sizes. My major concern is the organization and structure of the review, which is rather dense and descriptive and difficult to understand the key points. The main text needs to be checked extensively for grammatical errors and typos and missing citations throughout.

General Comments

Introduction

- There have been numerous reviews over the years focused on genetic and non-genetic factors in TS/CTD. How is this review different and what is its contribution? And what was the rationale for focusing on children and adolescents specifically? These points should be addressed in the introduction.

Methods

- Past tense (was) should be used throughout the methods.

- It is unclear why all versions of DSM were listed as inclusion criteria. It may be clearer to instead state that TS/CTD must be diagnosed based on established DSM criteria.

- A brief description of specific criteria assessed using the Newcastle-Ottawa scale would be useful.

Results

- Much of the written text in section 3.1 Characteristics of Studies could be instead reworked into the table instead of written out. For example, Table 1 could also include a column for “Study Population” to denote TS or CTD. The specific DSM version used for diagnosis is not relevant to report in my opinion. The populations are already listed in Table 1 and therefore can be removed from the main text.

- Similarly, the information in Table 2 could also be incorporated into Table 1 so that the study quality can be directly compared to other information about each study. Perhaps the “Study Quality” could be reported in the main table, and the individual NOS values could be included in supplemental as they do not really contribute to the overall conclusions.

- Once these information are combined into a single table, I would suggest separating out the studies by whether they focused on genetic factors vs non-genetic factors (environmental, psychological, etc.) for easy comparison within those topics.

- This section was difficult to read and interpret due to the volume of studies described and the specific statistics and samples sizes reported. This section would be clearer if it focused on the main findings and grouped similar studies together and provide summary statements of the key points of each section to guide the reader.

Discussion

- The discussion seems to reiterates the results from a higher level, which is helpful given that the “Results” section is rather dense with information and statistics about each specific study. However, the discussion should focus on summarizing the results, providing a synthesis of results across studies, and give clear recommendations for the future based on the findings. How might these studies be improved? And describe the specific implications of higher quality research on the understanding of TS/CTD and the diagnosis or treatment of these disorders.

- Experts are increasingly moving toward describing tic disorders as a spectrum rather than discrete diagnoses such as TS vs. CTD vs. provisional tic disorder, etc. There does not seem to be any discussion of differences between the risk of developing TS vs CTD and how factors may (or may not) differ between the two.

Minor Comments/Edits

- Are there supposed to be header in the abstract (Introduction, Methods, etc.)? If so they are difficult to pick out. This may be just how it is stylized in this form.

- The “Results” of the abstract could be clarified by focusing on the main factors with the most evidence (i.e., the factors highlighted in the “Conclusions” section) instead of listing all possible factors. Then the “Conclusions” section could focus more on future implications of understanding the pathogenesis of TS/CTD on diagnosis/treatment/etc.

- The manuscript needs to be checked extensively for grammar and typographical errors, as well as missing citations (e.g. page 13, line 448).

Author Response

Respose to the Review Comments

Dear reviewers and editors:

Thank you for your decision and constructive comments on my manuscript. We have carefully considered the suggestion of Reviewer and make some changes. We have tried our best to improve and made some changes in the manuscript.

Overview

Jiang et al present a review of case control studies focused on identifying genetic and non-genetic factors associated with the pathogenesis of Tourette syndrome (TS) and chronic tic disorder (CTD). The authors provide a summary of studies that also reports quality measures for each study. Although it is comprehensive, it does not clearly present a meaningful synthesis of results across studies or provide suggestions for future research other than the need for larger sample sizes. My major concern is the organization and structure of the review, which is rather dense and descriptive and difficult to understand the key points. The main text needs to be checked extensively for grammatical errors and typos and missing citations throughout.

General Comments

Introduction

- There have been numerous reviews over the years focused on genetic and non-genetic factors in TS/CTD. How is this review different and what is its contribution? And what was the rationale for focusing on children and adolescents specifically? These points should be addressed in the introduction.

Responds: Thank you for your suggestions. This review included not only studies written in English but only the studies written in Chinese which no wonder would enlarge the evidence for the etiology of TS/CTD. We had addressed those points in the introduction.

Methods

- Past tense (was) should be used throughout the methods.

- It is unclear why all versions of DSM were listed as inclusion criteria. It may be clearer to instead state that TS/CTD must be diagnosed based on established DSM criteria.

- A brief description of specific criteria assessed using the Newcastle-Ottawa scale would be useful.

Responds: Thanks for your comments

-We had revised the methods section, and used the past tense

-Because the diagnostic criteria of TS/CTD had changed over time, the studies in different time may according to the different versions of DSM, so that we list all versions of DSM. Different versions of the DSM had not made significant changes in the diagnostic criteria for TS/CTD.

-Thank you for your advice. We had given a brief description of specific criteria of Newcastle-Ottawa scale

Results

- Much of the written text in section 3.1 Characteristics of Studies could be instead reworked into the table instead of written out. For example, Table 1 could also include a column for “Study Population” to denote TS or CTD. The specific DSM version used for diagnosis is not relevant to report in my opinion. The populations are already listed in Table 1 and therefore can be removed from the main text.

- Similarly, the information in Table 2 could also be incorporated into Table 1 so that the study quality can be directly compared to other information about each study. Perhaps the “Study Quality” could be reported in the main table, and the individual NOS values could be included in supplemental as they do not really contribute to the overall conclusions.

- Once these information are combined into a single table, I would suggest separating out the studies by whether they focused on genetic factors vs non-genetic factors (environmental, psychological, etc.) for easy comparison within those topics.

- This section was difficult to read and interpret due to the volume of studies described and the specific statistics and samples sizes reported. This section would be clearer if it focused on the main findings and grouped similar studies together and provide summary statements of the key points of each section to guide the reader.

Responds: Thanks for your comments.

- The contents of Table 1 had been modified as required.

- The contents of Table 1 and Table 2 had been integrated.

Discussion

- The discussion seems to reiterates the results from a higher level, which is helpful given that the “Results” section is rather dense with information and statistics about each specific study. However, the discussion should focus on summarizing the results, providing a synthesis of results across studies, and give clear recommendations for the future based on the findings. How might these studies be improved? And describe the specific implications of higher quality research on the understanding of TS/CTD and the diagnosis or treatment of these disorders.

- Experts are increasingly moving toward describing tic disorders as a spectrum rather than discrete diagnoses such as TS vs. CTD vs. provisional tic disorder, etc. There does not seem to be any discussion of differences between the risk of developing TS vs CTD and how factors may (or may not) differ between the two.

Responds: Thanks for your comments.

- The contents of the discussion section had been revised as requested.

- We agree that tic disorders are a spectrum, but most studies have focused on CTD and TS, not individuals with PTD. Our study was not designed to distinguish between TS and CTD.

Minor Comments/Edits

- Are there supposed to be header in the abstract (Introduction, Methods, etc.)? If so they are difficult to pick out. This may be just how it is stylized in this form.

- The “Results” of the abstract could be clarified by focusing on the main factors with the most evidence (i.e., the factors highlighted in the “Conclusions” section) instead of listing all possible factors. Then the “Conclusions” section could focus more on future implications of understanding the pathogenesis of TS/CTD on diagnosis/treatment/etc.

- The manuscript needs to be checked extensively for grammar and typographical errors, as well as missing citations (e.g. page 13, line 448).

Responds: Thanks for your comments.

- The header had been highlighted.

- The sections of results and conclusions had been modified.

- We had tried our best made corrections for grammatical and typographical errors, and we also had asked someone who was good at English to help us polish it up.

Reviewer 2 Report

Article needs extensive editing by a proficient english speaker/writer. Many words seems to be missing the first letter. (eg. 'here' instead of 'there'). Grammatical errors (understood vs understand)

Although a review observational studies by it self is considered as a 'low quality' source of evidence, nevertheless these reviews are important in understanding the causes and conditions that cannot be studied under a controlled environment. The authors have done a commendable job in exploring the literature to identify the etio-pathogenesis and association of genetic factors for development TS/CTD. 

The review needs to be more focused on the available conclusions and the as the results does not seems to be reliable and in line with the review question.

A completed PRISMA checklist (available atprisma-statement.org/PRISMAStatement/Checklist.aspx) should be included for the reviewers and could be used as a guide for the authors to restructure the manuscript. 

Author Response

Respose to the Review Comments

Dear reviewers and editors:

Thank you for your decision and constructive comments on my manuscript. We have carefully considered the suggestion of Reviewer and make some changes. We have tried our best to improve and made some changes in the manuscript.

Article needs extensive editing by a proficient english speaker/writer. Many words seems to be missing the first letter. (eg. 'here' instead of 'there'). Grammatical errors (understood vs understand)

Responds: We had read through the article in detail and tried to revised all the grammatical and spell errors, and we also had asked someone who was good at English to help us polish it up.

Although a review observational studies by it self is considered as a 'low quality' source of evidence, nevertheless these reviews are important in understanding the causes and conditions that cannot be studied under a controlled environment. The authors have done a commendable job in exploring the literature to identify the etio-pathogenesis and association of genetic factors for development TS/CTD. 

The review needs to be more focused on the available conclusions and the as the results does not seems to be reliable and in line with the review question.

Responds: Thank you for your suggestions. We have made an effort to revise the article.

A completed PRISMA checklist (available atprisma-statement.org/PRISMAStatement/Checklist.aspx) should be included for the reviewers and could be used as a guide for the authors to restructure the manuscript. 

Responds: Thank you for your advice. We have We submitted a completed PRISMA checklist together this time.

Round 2

Reviewer 1 Report

Thank you to the authors for their revisions and replies to the review comments. The manuscript is somewhat improved, with a clearer presentation of the results in the table and some synthesis of the main findings in the discussion section. However, the contributions of this specific review to the field of TS research remain unclear to me; the authors note that it is the first to include summaries of Chinese studies. How do these studies that have not been previously summarized contribute to the overall conclusions? Are there case control studies in other languages besides English and Chinese that have not been included in this review or in other previous reviews? Additionally, I still urge the authors to check carefully for spelling and grammar errors in the table as well as the main text.

Author Response

Response to the comments

Dear reviewer:

Thank you for your decision and constructive comments on my manuscript. We have carefully considered the suggestion of the reviewer and made some changes. We have tried our best to improve and made some changes in the manuscript.

Thank you to the authors for their revisions and replies to the review comments. The manuscript is somewhat improved, with a clearer presentation of the results in the table and some synthesis of the main findings in the discussion section. However, the contributions of this specific review to the field of TS research remain unclear to me; the authors note that it is the first to include summaries of Chinese studies. How do these studies that have not been previously summarized contribute to the overall conclusions? Are there case control studies in other languages besides English and Chinese that have not been included in this review or in other previous reviews? Additionally, I still urge the authors to check carefully for spelling and grammar errors in the table as well as the main text.

Response: Thank you for your comments.

Firstly, the inclusion criteria we formulated at the beginning of the study had no language restrictions. Perhaps due to the limited databases retrieved, no studies in other languages that met the criteria were included in the study except those in English and Chinese. Through our review, we have added some evidence for the aetiology of TS/CTD, for example, MP infection, environmental factors, and psychological factors. In addition, we have made efforts to revise the grammar and spelling of the whole text.

Reviewer 2 Report

Minor spell check. T is still missing in some words. Line 10, 11, 40 .etc.

Minor grammatical issues including misplaced modifiers, unclear sentences still persist.

At its present form, this review covers all the basics and the conclusions are more focused, but requires a professional write up. 

Author Response

Response to the comments

Dear reviewer:

Thank you for your decision and constructive comments on my manuscript. We have carefully considered the suggestion of the reviewer and made some changes. We have tried our best to improve and made some changes in the manuscript.

Minor spell check. T is still missing in some words. Line 10, 11, 40 .etc.

Minor grammatical issues including misplaced modifiers, unclear sentences still persist.

At its present form, this review covers all the basics and the conclusions are more focused, but requires a professional write up. 

Response: Thank you for your comments.

We have checked the spelling of the whole text and corrected the errors.

We have checked the sentences of the whole text and corrected the inappropriate ones.

We have corrected the way the full text is written to make it more professional.
